# Private Data Leakage via Exploiting Access Patterns of Sparse Features in Deep Learning-based Recommendation Systems

**Hanieh Hashemi**[*]
University of Southern California
hashemis@usc.edu

**Wenjie Xiong**
Meta AI/Virginia Tech
wenjiex@meta.com

**Liu Ke**[*]
Washington University in St. Louis
ke.l@wustl.edu

**Kiwan Maeng**[*]
Pennsylvania State University
kvm6242@psu.edu

**Murali Annavaram**
University of Southern California
annavara@usc.edu

**G. Edward Suh**
Meta AI
edsuh@meta.com

**Hsien-Hsin S. Lee**[*]
Intel
lee.sean@gmail.com

## Abstract

Deep Learning-based Recommendation models use sparse and dense features of a user to predict an item that the user may like. These features carry the users' private information, service providers often protect these values by memory encryption (e.g., with hardware such as Intel's SGX). However, even with such protection, an attacker may still *learn information about which entry of the sparse feature is nonzero* through the embedding table access pattern. In this work, we show that only leaking the sparse features' nonzero entry positions can be a big threat to privacy. Using the embedding table access pattern, we show that it is possible to identify or re-identify a user, or extract sensitive attributes from a user. We subsequently show that applying a hash function to anonymize the access pattern cannot be a solution, as it can be reverse-engineered in many cases.

## 1 Introduction

Deep learning-based personalized recommendation models empower modern Internet services. These models exploit different types of information, including user attributes, user preferences, user behavior, social interaction, and other contextual information Erkin et al. (2010) to provide personalized recommendations relevant to a given user. They drive 35% of Amazon's revenue Gupta et al. (2020) and influence 80% of the videos streamed on Netflix Gomez-Uribe and Hunt (2015).

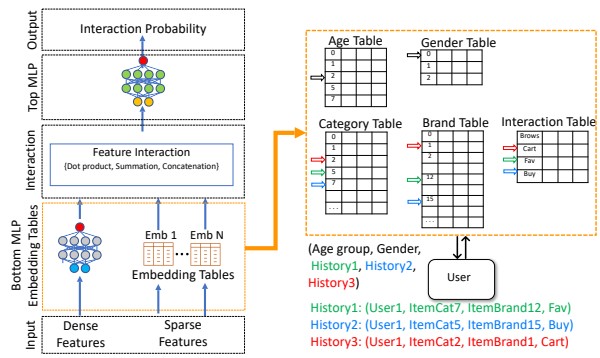

Figure 1: left: DLRM, right: example of embedding lookup.

---

[*]This work was conducted while the authors were employed at Meta.

2022 Trustworthy and Socially Responsible Machine Learning (TSRML 2022) co-located with NeurIPS 2022.

Deep learning-based recommendation models use dense (continuous) and sparse (categorical) features of a user as an input to a deep neural network to predict an item that a user may like (Figure 1, left). The features may include both static features that do not change frequently (e.g., age or gender) and dynamic features that changes frequently (e.g., a user's recent behavior history). Both features can hold sensitive information and must be kept private. Private user features are often encrypted in memory for privacy, using hardware such as trusted execution environment (TEE), e.g., Intel SGX team (2022). However, even when using hardware like TEE, the information of *which entries of the sparse features are nonzero* can be leaked. This is because sparse features must be projected into a lower-dimension space through an embedding table, where the index of the nonzero entries are used as an index for an embedding table lookup (Figure 1, right). In this paper, we show that *this information leakage can be an enough threat to privacy*. We first show that it is possible to (1) identify a user, (2) extract sensitive attributes of a user, or (3) re-identify a user, by only looking at the embedding table access pattern even when the data is fully encrypted. We subsequently show that applying a hash function to randomize the access pattern cannot be a general solution, by demonstrating a set of hash-inversion attacks. Specifically, we show that the below attacks are possible *by only observing the embedding table access patterns* in modern deep learning recommendation models:

- **Identification attack.** We demonstrate it is possible to identify a user by only observing the access pattern of sparse features' embedding table access pattern.
- **Sensitive attribute attack.** We show it is possible to extract sensitive attributes of a user (e.g., demographics) from seemingly unrelated sparse features, such as dynamic user behavior history.
- **Re-identification attack.** We show it is possible to identify if two queries are from the same user by only looking at seemingly innocuous sparse features, such as the users' recent purchase history.
- **Hash inversion with frequency-based attack.** We show that hiding the access using a hash cannot be a solution against these attacks, by demonstrating a hash inversion attack based on the access frequency. Our hash inversion attack can invert even sophisticated private hash functions as well as simple hash functions that are mainly used by the industry today.

## 2   Background and Threat Model

Deep learning-based recommendation model Zhou et al. (2018, 2019); Naumov et al. (2019); Ishkhanov et al. (2020); Cheng et al. (2016) uses dense and sparse features of a user and an item to predict whether the user will likely to interact with the item (e.g., click an Ad or purchase an item). Figure 1 shows the operation of a representative recommendation model, DLRM Naumov et al. (2019). In DLRM, the dense features go through a bottom MLP layer, while the sparse features go through an embedding table layer and get converted into a lower-dimensional dense features. Then, the two outputs go through a feature interaction layer (e.g., pairwise dot product) and go through a top MLP layer to predict the likelihood of an interaction. Other modern recommendation models work similarly Zhou et al. (2018, 2019); Ishkhanov et al. (2020); Cheng et al. (2016). Embedding tables convert a sparse feature into a dense representation by using the index of the nonzero entries in the sparse features as an index to perform lookup to a large table (Figure 1, right). Even when the entire dense and sparse features are fully encrypted and processed on a secure environment (e.g., by using Intel SGX Costan and Devadas (2016), hardware that encrypts content in the memory and protects computations), it is possible to learn which index holds a nonzero entry by looking at the table access pattern.

**Threat Model**   We assume a scenario where users share their private features with the service provider to get recommendations from the model. We assume that the values of the dense and sparse features of a user is fully protected from the attacker, e.g., with Intel SGX team (2022), but the *access pattern of the embedding table is revealed*, essentially revealing which entries are nonzero in the sparse features. In the real world, a honest-but-curious service provider running model inference on Intel SGX can fall into this category. Figure 2 demonstrates our threat model.

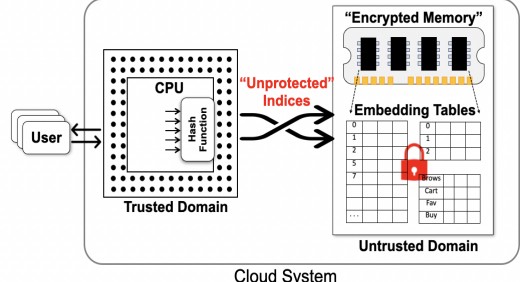

Figure 2: Our threat model assumes only the access pattern to the embedding table is revealed.

Table 1: Attack summary.

| Attack | Goal | Assumption | Evaluation Metric |
|---|---|---|---|
| Identification | Finding the identity of users | Attacker observes accesses
Has prior knowledge about distribution of accesses | K-anonymity |
| Sensitive Attribute | Extracting sensitive user features | Attacker observes accesses
Has prior knowledge about distribution of accesses | Ambiguity |
| Re-Identification | Tracking users over time | Attacker observes accesses | Precision and Recall |
| Frequency-based attack | Finding users' raw feature values | Attacker observes accesses
Has prior knowledge about distribution of accesses
Knows hash function
Does not know secret key for hash | Inversion Accuracy |
| OMP-based frequency attack for private hash | Finding users' raw feature values | Attacker observes accesses
Has prior knowledge about distribution of accesses | Inversion Accuracy |

Table 2: The number of users with anonymity level bellow K in the identification attacks (out of 1.14 million users).

| 1-anonymity | 2-anonymity | 3-anonymity | 4-anonymity | 5-anonymity | 6-anonymity | 7-anonymity | 8-anonymity | 9-anonymity | 10-anonymity |
|---|---|---|---|---|---|---|---|---|---|
| 56 | 154 | 256 | 380 | 480 | 606 | 739 | 867 | 984 | 1104 |

Table 1 summarizes the attacks we explored in this work. It includes other assumptions we had for each of the attacks and attacker's knowledge. We explained each attack in a separate section with details.

# 3 Identification Attack with Static User Features

A single user's inference request contains a series of sparse features, each of which in isolation has limited user information. However, multiple sparse features together can form a distinctive fingerprint for personal identification. User profile attributes (e.g. gender, city, etc) are usually static, in other words, they do not change or the frequency of the change is extremely low. We categorize this type of features into two subcategories—identifiable features and unidentifiable features. However, because of strict regulations in many domains, most of the recommendation systems do not collect and use such identifiable features. The question is if *unidentifiable* features such as age, gender, education, and shopping history can provide sufficient information to identify a user.

**Evaluation Setup:** To answer this question, we analyzed an open-source dataset released by Alibaba. This dataset contains static user features including user ID (1.14M), micro group ID (97), group ID (13), gender (2), age group (7), consumption grade/plevel (4), shopping depth (3), occupation/is college student (2), city level (5). More details about datasets is on Appendix A.

**Attack Method** In this set of features, the only directly identifying feature associated with a single user is the user ID. After removing the user ID, the collection of all other features provides 2.1 million possible combination. Hence, after removing the user ID, a user may mistakenly think that he or she is anonymous, and revealing any of the other features to the attacker on its own will not reveal the identity of the user. However, based on the user profile information from more than 1 million users, it is observed that in the real world only 1120 combinations of these static feature values are possible based on the real ope-source data. We refer to this 1120 as *user buckets*. We plotted the histogram of users in these 1120 buckets as shown in Figure 3. The

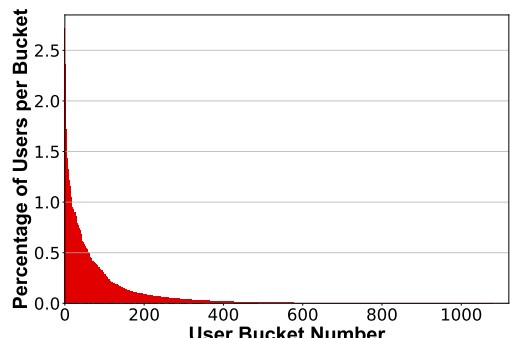

Figure 3: Percentage of the users belong to each user bucket.

x-axis in the figure indicates the bucket number ($[1 - 1120]$) and the y-axis shows the percentage of users per bucket. This histogram is quite illuminating in how the user distributions follow a long tail pattern. In particular, there are only a few users in buckets 600 to 1120. In fact, there are only 989 users on average across all these buckets, and the last 56 buckets have only 1 user. Consequently, observing the entire combinations of seemingly innocuous features from each allow may allow an attacker to launch an *identification attack* to extract the unique user ID with very high certainty.

**Evaluation Metric:** For our analysis, we used a well-known property known as *K-anonymity* used in information security/privacy. It describes a scenario in which if a user's bucket number is revealed and there are K users in the same bucket, the probability of finding the user is $\frac{1}{K}$. For instance,

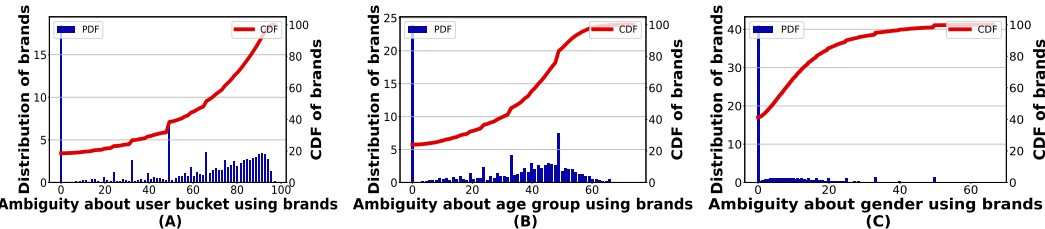

Figure 5: Using the accessed brands, ambiguity about A) user buckets (defined in previous section), B) user age groups, and C) user gender groups.

1-anonymity for a user means that this is the only user having this particular set of feature values.
**Evaluation Result:** As shown in Table 2, for $56$ of the user buckets, there is only one user with the specific combination of static features which implies that an attacker can identify these users with 1-anonymity if they can observe this combination of feature values. Also for more 1000 users, the anonymity level is 10 or below.

## 4 Sensitive Attribute Attack by Dynamic User Features

In this section, the question is when the user removes the static features, can sensitive features leak through other non-sensitive features? For instance, a user may provide no age information and they may have a sense of protecting more of their private data by not disclosing their static features. However, we demonstrate that even when a user hides their sensitive static features, adversaries are still able extract the sensitive attributes through cross correlations with user-item interaction data.

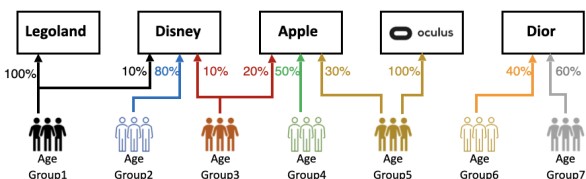

Figure 4: Different brands are popular between different customer age groups

**Evaluation Setup:** For evaluation, we use dynamic sparse features that includes user-item interactions Zhao et al. (2019) in the Alibaba Ads Display dataset. This dataset contains $723,268,134$ tuples collected over three weeks. Each tuple includes a user ID ($1.14M$), a btag (4: browse, cart, favor, buy), a category id ($12K$), and a brand ($379K$).

**Attack Method:** Figure 4 depicts an example of how different brands of the items are accessed by different user groups. The user/item interactions are depicted as graphs where each edge weight represents the fraction of the total interactions with that specific item from the corresponding age group. In real-world datasets, there are certain brands, where users from just a single age group interact with, in this example Legoland. A user who wants to protect their age group may not provide their age, but the adversary may deduce their age with a high probability if the user interacted with Legoland. While this simple illustration highlights the extremity (only one age group interacting with an item), this approach can be generalized. In General attacker, uses their prior knowledge on popularity of the items between different demographic groups. Then based on this prior information, they link the query to the demographic who formed most of the accesses to that item. Please note this prior information can be extracted by the users who are willing to share their information. Furthermore, for some of the products this is part of the product information.

**Evaluation Metric:** In this part, we employ a metric called *ambiguity* to determine the likelihood an adversary *fails* to predict a user's static sparse feature by just viewing their interactions with items. We define ambiguity for each item $i$ as: $ambiguity_i = 100\% - max(frequency_i)$ where $frequency_i$ is the distribution vector of all accesses to brand $i$ by different user groups. Using Figure 4 as an example, $frequency_{apple} = [0, 0, 20\%, 50\%, 30\%, 0, 0]$ and as a result $ambiguity_{Apple} = 50\%$, meaning if a user has interacted with item $i$ (Apple), the attacker can predict the static feature (age group) successfully for $50\%$ of the users. With this definition, $ambiguity_i = 0$ indicates if a user has interacted with item $i$, the attacker can successfully determine the user's sparse feature.

**Evaluation Result:** As shown in Figure 5, we quantify the ambiguity of predicting a user's sparse feature, such as age and gender, by using their item (brand) interaction history alone. The x-axis of these figures shows the percentage of ambiguity where a value of 0 indicates that there is no ambiguity,

and this brand is always accessed by only one user bucket. On the other hand, higher values indicate more ambiguity, and hence brands with higher values on the x-axis are popular across multiple user buckets. We plot both probability density function (PDF) and cumulative distribution function (CDF) of the ambiguity of different brands. What is revealing in the data is that in Figure 5(A), we observe that more than $17\%$ of brands are only accessed by 1 user bucket represented by the leftmost tall bar of PDF, meaning the attacker can determine the user bucket using those brands interactions. As shown in the CDF curve in Figure 5(A), for $38\%$ of the brands, the attacker can predict the user bucket with a success rate of greater than $50\%$. We present the information of age and gender group versus ambiguity in Figure 5(B) and Figure 5(C) respectively.

# 5 Re-Identification Attack

In re-identification attack, the goal of an attacker is to identify the same user over time by just observing their interaction history. Studies have shown the majority of the users prefer not to be tracked even anonymously Teltzrow and Kobsa (2004). Please note that this attack is different from identity resolution attack Bartunov et al. (2012), which tries to link the users whiten different systems. In this section, we first study if the history of the purchases of a user can be used as a tracking identifier for the user. Hence, we analyze if the history of the purchases is unique for each user. Second, we study if an attacker can re-identify the same user who sent queries over time by only tracking the history of their purchases, with no access to the static sparse features.

**Evaluation Setup:** For evaluation we used Taobao datase that has more than 723 million user-item interactions. Within them, we separated about 9 million purchase interactions. We then pre-processed and formatted that data in a time series data structure (*user history data structure*) shown below:

$$user_1 : (time_1, item_1), (time_4, item_{10}), (time_{500}, item_{20})$$
$$user_2 : (time_3, item_{100}), (time_{20}, item_{100})$$
$$\vdots$$
$$user_X : (time_5, item_{75}), (time_{20}, item_{50}),$$
$$(time_{100}, item_{75}), (time_{400}, item_1)(time_{420}, item_{10})$$

Second, for each set of consecutive items purchased by any user, we create a list of users who have the same set of consecutive purchases in exactly that order. We refer to these sets of consecutive recent purchases as **keys**. Multiple users may have the same key in their history. That is why each key keeps a *list* of all the users that created the same key and the duration of the time they had the key. An example of the *recent item purchase history* when we consider two most recent purchases shown below. Each key consists of a pair of items. For instance, the first line shows item 1 and item 10 were the most recent purchases of user 1 from time $4$ to time $500$.

$$key : \text{list of values}$$
$$[item_1, item_{10}] : [user_1, time_4, time_{500}]$$
$$[user_X, time_{420}, Current]$$
$$[item_{10}, item_{20}] : [user_1, time_{1000}, Current]$$
$$[item_{100}, item_{100}] : [user_2, time_{20}, Current]$$
$$\vdots$$
$$[item_{75}, item_{50}] : [user_X, time_{20}, time_{100}]$$
$$[item_{50}, item_{75}] : [user_X, time_{100}, time_{400}]$$
$$[item_{75}, item_1] : [user_X, time_{400}, time_{420}]$$

The goal of the this attack is to use only the $m$ ($m = 2$ in the example above) most recent purchases by a user to track the user across different interaction sessions, which are separated by timestamps as sessions. To evaluate this attack:

1. We randomly select a timestamp and a user.

2. For the selected user, we check the $m$ most recent purchases of the user at the selected timestamp and form a key = [recent purchase 1, recent purchase 2, ... recent purchase m]

3. We look up this key in the recent item purchase history dataset. If the same sequence of $m$ most recent items appear on another user at the same time window, this means these recent purchases are not unique for that specific user at that time and cannot be used as a fingerprint of a single user.

4. On the other hand, if the $m$ item purchase history only belongs to that specific user, the duration of the time in which this key forms the most recent purchases of the user is extracted.

5. This experiment is repeated for many random time stamps and users to obtain $200,000$ samples.

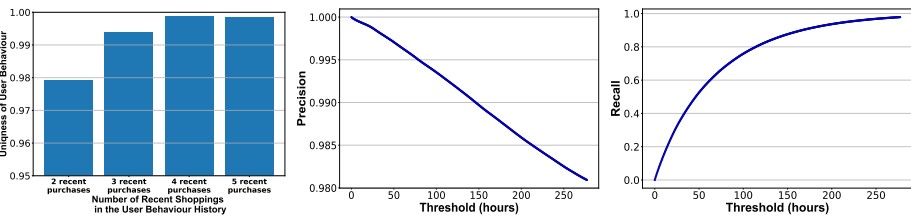

Figure 6: A) Uniqueness of most recent purchases of users. B and C) Precision/recall trade-off based on different time threshold values.

As depicted in Figure 6 A, we observe that even the two most recent purchases can serve as a unique identifier for $98\%$ of our samples. In other words, at a random point in time, the two most recent purchases of a user are unique for $98\%$ of randomly selected users. We found that three, four, and five most recent purchases uniquely identify users with $99\%$ probability.

**Attack Method:** Most recent items purchased by a user usually do not change with a very high frequency. For the period of time that these recent purchases remain the same, every query sent by the user has the same list of recent purchases. Therefore, the attacker is interested in using this knowledge to launch the attack. To accomplish this, the attacker first selects a time threshold. This time threshold is chosen to help the attacker to decide if the queries come from the same user or not. Meaning that if the time difference between receiving them is less than the time threshold and two distinct queries received by the cloud have the same most recent purchases, the attacker will predict that they comes from the same use. Otherwise, it is assumed queries come from two different users.

**Evaluation Metric:** To measure the accuracy of this attack, we use the machine learning terms *precision* and *recall* defined in Buckland and Gey (1994) as shown in Eq (1).

$$Precision = \frac{TP}{(TP + FP)}, \quad Recall = \frac{TP}{(TP + FN)} \; , \tag{1}$$

where TP stands for **T**rue **P**ositives, FP represents **F**alse **P**ositives, and FN is **F**alse **N**egatives. Precision indicates what percentage of positive predictions are accurate and Recall indicates what percentage of actual positives are detected.

**Evaluation Result:** To evaluate the precision/recall tradeoff, we start from a very small time threshold and increase it gradually. As expected, with low time thresholds, precision is high with few false positives. But as the attacker increases the time threshold and can identify more of the actual positives (higher recall), they false positives increase as well, which reduces the precision. The reason for having more false positives with a large threshold is that, during a longer period of time, other users may generate the same key. Table 3 shows when the 2 most recent purchases are used, there are around $4.5$ million keys but the total number of occurrences of these keys is around $8$ million times. This means for a fraction of the keys, the same keys are generated for different users at different times. These repeated keys are the source of false positives in our experiments. The decision of selecting the right threshold depends on the attacker's preference to have a higher recall or precision.

Figure 6 shows this trade-off for different time threshold values. We gradually increase the time threshold from 1 second to 277 hours (11.5 days). As shown in this figure, by increasing the time threshold to 11 days recall will reach $1.0$ while there is an almost $0.02$ drop in precision. This means the attacker can link all the queries that come from the same users correctly. This comes at the cost of $2\%$ miss-prediction of the queries that do not come from the same user and only generates the same key at some point in their purchase history. *These high precision and recall values, indicates how an attacker can track users who send queries to the recommendation model over time.*

Table 3: Re-identification attack statistics about the number of keys and repeated keys.

| Number of recent purchases | Number of users | Number of keys | Total occurrences of keys |
|---|---|---|---|
| 2 | $898,803$ | $4,476,760$ | $8,114,860$ |
| 3 | $799,475$ | $5,679,087$ | $7,216,057$ |
| 4 | $705,888$ | $5,587,578$ | $6,416,582$ |
| 5 | $620,029$ | $5,197,043$ | $5,710,694$ |

# 6 Hash inversion with frequency-based attack

Applying hash on the indices before embedding table lookup is an important performance optimization (more details about the data pipeline in production-scale recommendation systems and different hashing schemes can be found in Appendix B). Here, we analyze how hashing impact information leakage. This section studies how an attacker can recover the raw values of sparse features even when hashing is used for embedding indices. Through a hash function, users' raw data are remapped to post-hash values for indexing the embedding tables as shown in Fig. 7.

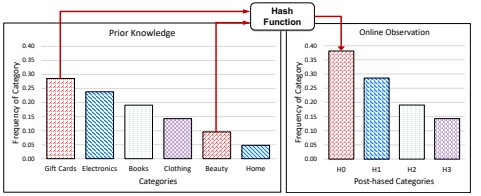

Figure 7: Frequency-based attack tries to reverse engineers the hash based on the frequencies.

**Evaluation Setup:** For evaluation, we used Taobao, Kaggle and Criteo datasets. For each dataset we selected two disjoint random sets; training set and test test. The training set samples forms the prior distribution and the test sample are used for the evaluation.

**Attack Method:** An adversary can launch attacks by collecting the frequency of observed indices,

Table 4: Accuracy of hash inversion for the frequency-based attack for Taobao dataset.

| Number of Samples used for Learning Distribution | Number of Samples for Evaluation | Top 1 | Top 2 | Top 3 | Top 4 | Top 5 | Top 6 | Top 7 | Top 8 | Top 9 | Top 10 |
|---|---|---|---|---|---|---|---|---|---|---|---|
| 1,000,000 | 1,000 | 0.64 | 0.76 | 0.83 | 0.87 | 0.89 | 0.90 | 0.91 | 0.92 | 0.93 | 0.94 |
| 1,000,000 | 100,000 | 0.61 | 0.75 | 0.82 | 0.86 | 0.88 | 0.90 | 0.92 | 0.92 | 0.93 | 0.93 |
| 2,000,000 | 100,000 | 0.62 | 0.76 | 0.82 | 0.86 | 0.89 | 0.91 | 0.92 | 0.93 | 0.93 | 0.94 |
| 2,000,000 | 1,000,000 | 0.62 | 0.76 | 0.82 | 0.86 | 0.89 | 0.91 | 0.92 | 0.93 | 0.93 | 0.94 |

use prior knowledge about the distribution of feature values, and find the mapping between input and output of the hash. Here we show how an attacker can compromise a system with hashed input values where the hash function is $output = (input + mask_{add}) \bmod P$ and $P$ is the hash size. We denote the frequency of possible input to a hash function by $x_1, x_2, \ldots, x_N$ for N possible scenarios and its output frequency by $y_1, y_2, \ldots, y_P$ of a hash size P. We form the matrix $M \in \mathbb{R}^{P \times P}$ in which each column represents a different value for Mask ($[0, P-1]$). Basically, for each value of a mask, we compute the frequency of outcomes and form this Matrix. As shown, by increasing the value of the mask by 1, the column values are shifted. Hence, the Matrix M is a Toeplitz Matrix. Since a single column in this matrix is shifted and repeated the order of forming this matrix is $O(P)$.

$$\mathbf{M} = \begin{bmatrix} y_1 & y_{P-1} & \cdots & y_2 \\ y_2 & y_1 & \cdots & y_3 \\ \vdots & \vdots & \ddots & \vdots \\ y_P & y_{P-2} & \cdots & y_1 \end{bmatrix}_{P \times P} \tag{2}$$

The attacker's goal here is to invert the hash using the input distribution and its observation of the output distribution. Note an input dataset and an output dataset should be independent. We define $\mathbf{a}_t$ as the distribution of embedding table accesses (post-hash) at time t. To reverse engineer the mask, an attacker has to find out which mask is used by the hash function. To do so, the attacker has to solve the optimization problem in Eq( 3).

$$\min_i \quad \|(\mathbf{m}_i - \mathbf{a}_t)\|^2 = \min_i (\|\mathbf{m}_i\|^2 + \|\mathbf{a}_t\|^2 - 2\mathbf{m}_i^\mathsf{T}\mathbf{a}_t) \tag{3}$$

In Eq (3), $\mathbf{m}_i$ represents the vector containing the frequencies of output values when mask $i$ is used. So its absolute value will be a constant one. This is similar for $\|\mathbf{a}_t\|$. As a result, the optimization problem can be simplified to Eq(4).

$$\bar{P} = \arg\max_i(\mathbf{m_i}^\mathsf{T}\mathbf{a}_t) \quad for \quad i \in [0, P-1] \implies \bar{P} = \arg\max_i(\mathbf{M}^\mathsf{T}\mathbf{a}_t) \tag{4}$$

The order of computing such a matrix-vector product is $O(P^2)$. However, because $\mathbf{M}$ is a Toeplitz matrix, this matrix vector computation can be done in time complexity of $O(P \log P)$ Strang (1986). To implement this attack, we created two disjoint sets. The first set is used to extract the distribution (known distribution) and the second set is used for frequency matching and evaluating the frequency-based attack. First, attackers try to reverse engineers the hash function and find the key based on the frequency matching. The attacker was able to reverse engineer the hash and find the key based on the method described above. Next, the attacker tries to reverse engineer the post-hash indices and find out the value of raw sparse features. After finding the key of the hash, the attacker reverse engineer

the post-hash value to the top most frequent pre-hash values based on the input distributions.

**Evaluation Metric:** Accuracy in this case is the probability that the attacker correctly identifies an input raw value from the post-hash value. Let the function $g(y)$ be the attacker's estimate of the input, given the output query $y$, $g(y) = \arg\max_x \text{Prob}(x)$   s.t.   $\hat{h}(x) = y$ , where $\hat{h}(x)$ is the attackers estimation of the hash function. Using this definition, accuracy is defined:

$$\text{Accuracy} = \text{Prob}_{x \sim \mathcal{P}_X} \left( x = g(h(x)) \right) , \tag{5}$$

where $h(x)$ is the true hash function, and the probability is over the distribution of the input query. We also use the notation of *top K accuracy* in this section. Essentially top $K$ accuracy is the probability of the input query being among the top guesses of the attacker. To formally define this, we first denote the set $\hat{\mathcal{S}}(y)$ as, $\hat{\mathcal{S}}(y) = \{x \mid \hat{h}(x) = y\}$ , which is the set of all possible inputs, given an output query $y$, based on attacker's estimation of the hash function. We now define the set $g_K(y)$ to be the top $k$ members of the set $\hat{\mathcal{S}}(y)$ with the largest probability, $g_K(y) = \{x \in \hat{\mathcal{S}}(y) | \text{Prob}(x) \text{ is in the top } K \text{ probabilities.}\}$. This means that $g_K(y)$ is the set of the top $K$ attacker's guesses, of the input query. Now we can use the function $g_k(y)$ to formally define the top $K$ accuracy,

$$\text{Accuracy}_{\text{top } K} = \text{Prob}_{x \sim \mathcal{P}_X} \left( x \in g_K(h(x)) \right) , \tag{6}$$

where $h(x)$ is the true hash function, and the probability is over the distribution of the input query.

**Evaluation Result:** As shown in Table 4, we change the number of interactions in these test sets to see the accuracy of hash-inversion and the attacker could achieve up to $0.94$ top 10 accuracy for the Taobao dataset. Results on Kaggle and Criteo datasets are reported in C.*The key observation here is that, if an attacker observes the frequency of queries, they can reconstruct the values of raw features with high accuracy by knowing the distributions of the pre-hash values and type of the hash function.* We also expand this attack and support a general attack for more complex hash functions using OMP. The details of this machine learning based attack is explained in Appendix D. In Appendix F we disccussed why none of the current solutions can solve all the issues.

## 7   Potential Solutions

One approach to obfuscating the embedded table access pattern is to use Oblivious RAM (ORAM) Goldreich and Ostrovsky (1996); Stefanov et al. (2018); Ren et al. (2014). In a high level, for each read or write operation, ORAM controller reads and writes not only the requested block, but also many random blocks. In this way, ORAM hides the information about real blocks from the attacker. However, the overhead of ORAM is unlikely to be acceptable for real-time applications such as recommendation system inference due to Service Level Agreement (SLA) Hazelwood et al. (2018). Even the most optimized version of ORAM suffers from 8-10 times performance overhead Raoufi et al. (2022). A previous study Rajat et al. (2021) tries to optimize ORAM for recommendation systems training. But, the scheme relied on pre-determined sequence of accesses in training and is not applicable to inference. In our future work, we plan to investigate low-latency protection schemes for embedding table accesses in recommendation system inference.

## 8   Conclusion

In this work, we shed light on the information leakage through sparse features in deep learning-based recommendation systems. Our work pivoted the prior investigation focus on dense feature protection to the unprotected access patterns of sparse features. The new insight from this work demonstrates even the access patterns can be a big threat to privacy.

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

# A    Data sets

For studying the attacks in the following sections, we use multiple open source datasets such as Taobao Ads Display, Kaggle Ads Display, and Criteo Display. In this section, we briefly explain the content of these datasets, and in each of the following sections, we explain more about the dataset characteristics that we used.

**Taobao Ads Display Team (2018)**: This dataset contains user static features that includes $1,140,000$ users and 10 static features per user including their user IDs. There are also other features representing a user's profile, *e.g.*, age, gender, occupation level, living city, education level, etc. Another file contains user behavior data that includes seven hundred million records of user past behaviors. It contains shopping behavior over 22 days. Each row of this file indicates an interaction between a user (represented by user ID) and an item (represented by item brand ID and category ID). The type of interaction (buy, brows, fav, cart) and the time stamp of the interactions.

**Kaggle Ads Display Lab (2018b)**: CriteoLabs shared a week's worth of data for you to develop models predicting ads' click-through rates (CTR). This dataset contains three data files including

training file and test files. Training file consists of a portion of Criteo's traffic over a period of 7 days. Each row corresponds to a display ad served by Criteo. Positive (clicked) and negative (non-clicked) examples have both been subsampled at different rates to reduce the dataset size. Each row contains 13 dense features and 26 sparse features that form embedding table accesses. The semantic of these features is not released. The test set is computed in the same way as the training set but for events on the day following the training period.

**Criteo Ads Display Lab (2018a)**: This dataset is similar to Kaggle. But it is a much larger dataset containing 24 data files collected over 24 days with a different subsampling ratio.

For the identification attack, sensitive attribute attack, re-identification attack, and OMP-based frequency attack our analysis requires user IDs, static profile features, or user past behaviors in the same dataset. Hence, for these attacks, we used the Taobao dataset, which is the only public dataset containing all these features. For the frequency-based attack, we need less information to implement the attacks. Thus all the datasets meet the requirement and we evaluate all of them in the hash information leakage study and the frequency based attack.

# B  Data Pipeline in Production-Scale Recommendation Systems

As mentioned earlier, exposing raw values of sparse features can leak sensitive information of a user. In this section, we discuss the current production-scale data pipeline for sparse feature processing and how such real system designs may impact the information leak.

One challenge in designing efficient embedding tables is that the values of sparse features may be unbounded, resulting in very large embedding table sizes. Consider the news articles produced in the world as a dynamic sparse feature item that a user may interact with. There are thousands of news articles in just a day from around the world and creating embeddings for each news item in an embedding table is impractically large. For instance, the DLRM recommendation model in 2021 needs 16x larger memory, compared to the one used in 2017 Lui et al. (2021); Sethi et al. (2022). Furthermore, 99% of model parameters belong to embedding tables Gupta et al. (2020). That is why production-scale models demand 10s of TB memory capacity Mudigere et al. (2021); Sethi et al. (2022). One common solution for converting high dimensional data to a low-level representation is to use hashing Shi et al. (2009). Using hashing for recommendation systems was first suggested in Zhang et al. (2018). In addition to bounding sparse features to a fixed size, hashing helps with responding to the rare inputs that are not seen before Acun et al. (2021); Kang et al. (2020). Furthermore, using high-cardinality features may cause over-fitting problems due to over parameterization Liu et al. (2020); Kang et al. (2020). Considering all these reasons, sparse feature inputs in production-scale models are hashed prior to embedding look-ups.

In the appendix B.1, we briefly explain how different hashing schemes work and then we analyze how hashing impact information leakage. Recall that all the information leakage that we discussed in the prior sections is due to the fact that an adversary sees the raw value of embedding table indices. We analyzed and demonstrate embedding table hashing in recommendation systems, which was not necessarily designed for protecting data privacy could not help with reducing information leakage.

## B.1  Hash Functions

There are multiple ways of reducing the embedding table size using hash functions, and they all have trade-offs. We explain some of the most common hashing schemes here.

**Embedding table as a hash-map**: With hash-map, embedding table entries are combined based on their similarity and a smaller embedding table is formed. However, to use the embedding table, a hash map should be kept to keep track of merged entries. This is the most accurate but the most expensive method in practice. In a previous study Zhang et al. (2018), the authors suggested that using locality sensitive hashing can approximately preserve similarities of data while significantly reducing data dimensions. Frequency hashing Zhang et al. (2020) also keeps a separate map with hot items and carefully maps only hot items to different entries in the table. This ensures that hot items do not collide, while items that are less frequently accessed may in fact be mapped to a same entry.
**Modulo hashing**: This is the cheapest and simplest hash to implement. This hashing performs modulo division based on the pre-defined size of the hash table. For hash size $P$, the hash function is as simple as $input \ mod \ P$. Though simple, it has the disadvantage that two completely different

entities might collide.

**Cryptographic hashing**: This approach is a one-way cryptographic algorithm that maps an input of any size to a unique output of a fixed length of bits. A small change in the input drastically changes the output. Cryptographic hashing is a deterministic hashing mechanism.

## B.2    Statistical Analysis on Information Leakage After Hashing

In this section, we analyze if the amount of randomization created by hashing can have any effect on reducing data leakage. In the following, we report our analysis on the entropy of pre-hash and

Table 5: Entropy and mutual information analysis of pre-hash and post-hash embedding table indices.

| Dataset | Table Name | Original Table Size | Post Hash Table Size | Pre-Hash Entropy | Post-Hash Entropy | MI |
|---------|-----------|--------------------|--------------------|-----------------|------------------|------|
| Taobao | Brands | $379,353$ | $37,935$ | 9.91 | 9.28 | 9.28 |
| Taobao | Categories | $12,124$ | $1,212$ | 6.19 | 5.72 | 5.72 |
| Kaggle | C3 | $1,761,917$ | $176,191$ | 10.15 | 9.41 | 9.41 |
| Kaggle | C18 | $4,836$ | $483$ | 5.92 | 5.27 | 5.27 |
| Kaggle | C24 | $110,946$ | $11,094$ | 6.57 | 6.28 | 6.28 |
| Criteo | C7 | $6,593$ | $659$ | 7.63 | 5.84 | 5.84 |
| Criteo | C12 | $159,619$ | $15,961$ | 7.20 | 6.85 | 6.58 |
| Criteo | C20 | $11,568,963$ | $1,156,896$ | 7.37 | 7.18 | 7.18 |

post-hash indices as well as the mutual information analysis. Given a discrete random variable X, with possible outcomes: $x_1, \ldots, x_n$ which occur with probability $p(x_1), \ldots, p(x_n)$, the entropy is formally is defined as Cover (1999):

$$H(X) = -\sum_{i=1}^{N} p(x_i) \times log(p(x_i)) \tag{7}$$

The binary (Base 2) logarithm gives the unit of bits (or "shannons"). Entropy is often roughly used as a measure of unpredictability. In this part we measure the entropy of the input and output of the hash function. In our specific evaluation, we first measure the probabilities in Eq (7) by measuring the frequency of each outcome for pre-hash. We used modulo hash function for compressing the values and measured the post-hash frequencies. Finally by applying Eq (7), we find out the amount of uncertainty in each of these values. As shown in Table 5, the pre-hash entropy of the brand table in Taobao dataset is almost 10 bits. Even after reducing the table size with hashing by 10 times, the amount of information is not reduced significantly for the post-hash values. For the category table, the amount of information was 6 bits and it remains the same after 10 times reduction in the table size. For Kaggle, we selected three embedding tables with different sizes. C3 is the largest embedding table with $1,761,917$ entries. C18 represents the small tables with $4,836$ entries while C24 represents the moderate tables with $110,946$ entries. As shown in this table, the entropy of the sparse features varies between 10 bits to 6 bits depends on the feature. This entropy is not reduced significantly in the post hash values. Finally, the Criteo dataset is evaluated. Note that since the dataset is hashed in a different way, feature names are different from the Kaggle dataset. In this dataset, C7 is the smallest table with $6,593$ entries. C12 is the average-size table and C20 is the largest embedding table with $159,619$ and $11,568,963$ entries respectively. The details about embedding table sizes are reported in Appendix A. *An important observation is that the entropy of information in indices is not reduced significantly after hashing. It implies that the post-hash indices hold almost the same amount of information as the pre-hash indices.*

**Mutual Information (MI) Analysis** In probability and information theory, the mutual information of two random variables is a measure of the mutual dependence between the two variables. More specifically, it quantifies the "amount of information" obtained about one random variable by observing the other random variable. Mutual information between two random variables X and Y is measured by Cover (1999):

$$I(X;Y) = H(X) - H(X|Y) = H(Y) - H(Y|X) \tag{8}$$

Many prior works used MI as a measure of privacy guarantee Cuff and Yu (2016); Kalantari et al. (2017); Liao et al. (2017); Guo et al. (2020); Mireshghallah et al. (2020). In our example, we compute the mutual information between the pre-hash indices ($X$) and the post-hash indices ($Y$). Based on Eq(8), the mutual information between post-hash and pre-hash indices is equal to the entropy of the post-hash indices (H(Y)) minus the conditional entropy of post-hash indices given the pre-hash

indices ($H(Y|X)$). With deterministic hash functions, a post-hash index is deterministic for a given pre-hash index. This means there is no ambiguity in the conditional entropy. So $H(Y|X)$ in Eq( 7) is equal to zero and MI is equal to the entropy of post-hash indices. Our empirical result in Table 5 also validates this point. *Based on this observation, the mutual information between input and output of the hash is almost equal to the entropy of the hash input. This means that an adversary with unlimited computational power can recover almost all the information in the pre-hash indices by just observing the post-hash indices.*

## C   Frequency Based Attack: Kaggle and Criteo Datasets

In Table 6, we show the accuracy of this attack model for the Kaggle dataset. As demonstrated in this table for small embedding tables (represented by C18), even a small sample of prior distribution and online queries observed by an attacker can lead to a high inversion accuracy while for large tables (represented by C3) more accurate distributions are needed. The evaluation for the Criteo dataset is reported in Table 7. In this dataset C7 is the smallest table, C20 is the average-size table and C12 is the largest embedding table (More details about embedding table sizes are reported in Appendix A.). Criteo dataset also validates the same observation as previous datasets.

Table 6: Accuracy of hash inversion for the frequency-based attack for Kaggle dataset.

| Number of Samples used for Learning Distribution | Number of Samples for Evaluation | Feature | Top 1 | Top 2 | Top 3 | Top 4 | Top 5 | Top 6 | Top 7 | Top 8 | Top 9 | Top 10 |
|---|---|---|---|---|---|---|---|---|---|---|---|---|
| 100,000 | 1,000 | C3 | 0.55 | 0.55 | 0.55 | 0.55 | 0.55 | 0.55 | 0.55 | 0.55 | 0.55 | 0.55 |
| 100,000 | 1,000 | C18 | 0.74 | 0.90 | 0.95 | 0.96 | 0.98 | 0.98 | 0.98 | 0.98 | 0.98 | 0.98 |
| 100,000 | 1,000 | C24 | 0.87 | 0.92 | 0.92 | 0.92 | 0.93 | 0.93 | 0.93 | 0.93 | 093 | 0.93 |
| 1000,000 | 10,000 | C3 | 0.63 | 0.64 | 0.65 | 0.65 | 0.65 | 0.65 | 0.65 | 0.65 | 0.65 | 0.65 |
| 1000,000 | 10,000 | C18 | 0.75 | 0.89 | 0.94 | 0.96 | 0.98 | 0.98 | 0.98 | 0.99 | 0.99 | 0.99 |
| 1000,000 | 10,000 | C24 | 0.90 | 0.95 | 0.96 | 0.97 | 0.97 | 0.97 | 0.97 | 0.97 | 097 | 0.97 |
| 4,000,000 | 100,000 | C3 | 0.68 | 0.71 | 0.71 | 0.72 | 0.72 | 0.73 | 0.73 | 0.73 | 0.74 | 0.74 |
| 4,000,000 | 100,000 | C18 | 0.78 | 0.91 | 0.95 | 0.97 | 0.98 | 0.99 | 0.99 | 0.99 | 0.99 | 0.99 |
| 4,000,000 | 100,000 | C24 | 0.91 | 0.95 | 0.97 | 0.97 | 0.98 | 0.98 | 0.98 | 0.98 | 0.98 | 0.98 |

Table 7: Accuracy of hash inversion for the frequency-based attack for Criteo dataset.

| Number of Samples used for Learning Distribution | Number of Samples for Evaluation | Feature | Top 1 | Top 2 | Top 3 | Top 4 | Top 5 | Top 6 | Top 7 | Top 8 | Top 9 | Top 10 |
|---|---|---|---|---|---|---|---|---|---|---|---|---|
| 3,000,000 | 200,000 | C7 | 0.33 | 0.48 | 0.61 | 0.68 | 0.74 | 0.80 | 0.84 | 0.88 | 0.91 | 0.93 |
| 3,000,000 | 200,000 | C12 | 0.89 | 0.96 | 0.98 | 0.99 | 0.99 | 0.99 | 0.99 | 0.99 | 0.99 | 0.99 |
| 3,000,000 | 200,000 | C20 | 0.93 | 0.98 | 0.99 | 0.99 | 1 | 1 | 1 | 1 | 1 | 1 |
| 30,000,000 | 2,000,000 | C7 | 0.33 | 0.48 | 0.58 | 0.65 | 0.73 | 0.80 | 0.85 | 0.88 | 0.92 | 0.93 |
| 30,000,000 | 2,000,000 | C12 | 0.89 | 0.96 | 0.98 | 0.98 | 0.99 | 0.99 | 0.99 | 0.99 | 0.99 | 0.99 |
| 30,000,000 | 2,000,000 | C20 | 0.85 | 0.88 | 0.91 | 0.94 | 0.96 | 0.98 | 0.99 | 0.99 | 0.99 | 0.99 |
| 400,000,000 | 4,000,000 | C7 | 0.33 | 0.48 | 0.58 | 0.65 | 0.73 | 0.80 | 0.83 | 0.88 | 0.90 | 0.93 |
| 400,000,000 | 4,000,000 | C12 | 0.89 | 0.96 | 0.98 | 0.98 | 0.99 | 0.99 | 0.99 | 0.99 | 0.99 | 0.99 |
| 400,000,000 | 4,000,000 | C20 | 0.84 | 0.88 | 0.90 | 0.92 | 0.95 | 0.97 | 0.98 | 0.99 | 0.99 | 0.99 |

## D   Is Private Hash a Solution?

Note that hash functions are currently used for reducing the sizes of embedding tables rather than designed for privacy purposes. But if a private hash function is employed, can it guarantee zero information leakage? In other words, using any random mapping between inputs and outputs of the hash, and if an attacker does not know the hash, can they find the mapping just by observing the frequency of the accesses? To answer this question, we first use a simple greedy attack to demonstrate the leakage of information. Then we use a more sophisticated machine learning based optimization exploiting sequences of access to show how an attacker can achieve a high hash inversion accuracy even when the hash function is unknown.

We first design a greedy attack to map the inputs and outputs by matching the frequencies without having any further information about the hash function. The only knowledge the attacker has are the prior distribution of pre-hash accesses and the observed post-hash access to the embedding table. We analyzed the category table of $12,000+$ pre-hash entries and $1,200$ post-hash entries ($P = 0.1N$). We randomly map each of the $12,000$ inputs to an output. Then we launched the frequency-based attack without providing any information about this mapping to the attacker. This simple attack could successfully figure out the correct mapping for $23\%$ of the accesses. This analysis showed

that although a private hash can reduce the amount of information leakage, it will not eliminate the leakage completely and is still susceptible to this type of attack. Now we take a step further to show how this attack can achieve an even higher inversion accuracy.

**Evaluation Setup:** As we explained in the previous sections, the user shares their most recent behaviors with the recommendation system to receive accurate suggestions. In this section, we show that the combination of the users' past shopping behaviors within one query, can help attackers launch more sophisticated attacks. Hence, for evaluating this attack we use Taobao dataset that provides this shopping behaviours. We evaluated both Category and Brand tables with more than 379K and 12K raw entries respectively.

**Attack Method:** Assume that $N$ is the size of the input, and $P$ is the size of the output, and the hash function $\mathbf{h}(.)$ maps the input to the output. Thus, $\mathbf{h}[i] = j$ means that the hash function, maps input index $i$ to output index $j$. We do not impose any assumptions on the hash function in this part. Assume that the joint distribution of the indices of the input and the output are shown by the matrices $\mathbf{X} \in \mathbb{R}^{N \times N}$ and $\mathbf{Y} \in \mathbb{R}^{P \times P}$, respectively. This means that the probability of $(i_1, i_2)$ in the input is $\mathbf{X}_{i1,i2}$ and the probability of $(j_1, j_2)$ in the output is $\mathbf{Y}_{j1,j2}$. Also assume that the matrix $\mathbf{B} \in \mathbf{R}^{P \times N}$ is the one-hot representation of the hash function $\mathbf{h}(.)$, such that

$$\mathbf{B}_{j,i} = \begin{cases} 1 & \mathbf{h}(i) = j \\ 0 & \text{otherwise} \end{cases} \tag{9}$$

Using these notations, we can show that,

$$\mathbf{Y} = \mathbf{BXB}^T . \tag{10}$$

To prove this, note that

$$\mathbf{Y}_{i_1,i_2} = \sum_{j_1,j_2} \mathbb{1}_{\mathbf{h}(j_1)=i_1} \mathbb{1}_{\mathbf{h}(j_2)=i_2} \mathbf{X}_{j_1,j_2}$$
$$= \sum_{j_1,j_2} \mathbf{B}_{i_1,j_1} \mathbf{X}_{j_1,j_2} \mathbf{B}_{j_2,i_2} , \tag{11}$$

where $\mathbb{1}_{\mathcal{E}}$ is the indicator function of the event $\mathcal{E}$, therefore $\mathbb{1}_{\mathbf{h}(j_1)=i_1} = \mathbf{B}_{i_1,j_1}$. Eq (11) yields (10). Now, to estimate $\mathbf{B}$, we would like to ideally solve the following optimization.

$$\hat{\mathbf{B}} = \arg \min_{\mathbf{B} \in \mathcal{B}} \|\mathbf{Y} - \mathbf{BXB}^T\|_F^2 , \tag{12}$$

where $\|\mathbf{X}\|_F^2 = \sum_{i,j} \mathbf{X}_{i,j}^2$ is the Frobenius norm and $\mathcal{B}$ is the space of all possible matrices $\mathbf{B}$, that represents a hash function. Optimization (12) is an integer programming and NP-hard problem, due to the constraint in the minimization. To approximately solve this, we use Orthogonal Matching Pursuit (OMP) Tropp and Gilbert (2007). The idea behind OMP is to find one column of the matrix $\mathbf{B}$ in each iteration, in such a way that the new column satisfies the constraint on $\mathbf{B}$, and the new added column minimizes the loss function in (12) the most (compared to any other feasible column). Note that in each iteration of our algorithm, we make sure that the matrix $\mathbf{B}$ can represent a hash function. The size of Matrix $\mathbf{B}$ can grow large based on the embedding table size. Thus, in our implementation we used CSR format since this matrix is sparse.

**Evaluation Metric:** Accuracy is the probability that the attacker correctly identifies a raw input value from the post-hash value. We used top-1 accuracy which is defined in Eq (5).

**Evaluation Result:** To evaluate this attack, we measure the accuracy of the hash inversion function when changing the hash size. Figure 8 demonstrates the hash-inversion accuracy using this optimization for the Taobao category table. We used different hash sizes to evaluate this attack. The size of the hash table changes from $0.05$ ($P = 0.05N$) of the original table size to $0.80$ of the table size. It shows how this accuracy increases over iterations until it saturates. For the large hash sizes, $P = 0.8N$, accuracy reaches $94\%$, which means the this attack can recover raw values from hashed values for $94\%$ of accesses. Since the embedding table size for the Brand table is large, we used the Compressed Sparse Row (CSR) implementation to optimize the memory usage of the attacker. This way we could analyze the same attack on the brand embedding table with $379, 353$ raw entries. Figure 9 shows how different hash sizes can change the attacker's accuracy for hash inversion in the brand table. *The key takeaway is that, even an unknown private hash cannot reduce the information leakage. An attacker can use this frequency-based machine learning optimization to recover the raw value features with high accuracy.*

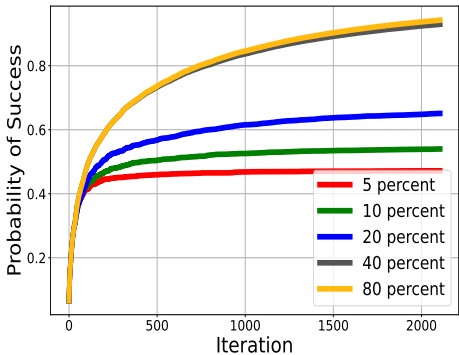

Figure 8: Hash-inversion accuracy increases with more optimization iterations and Larger hash sizes (Category Table).

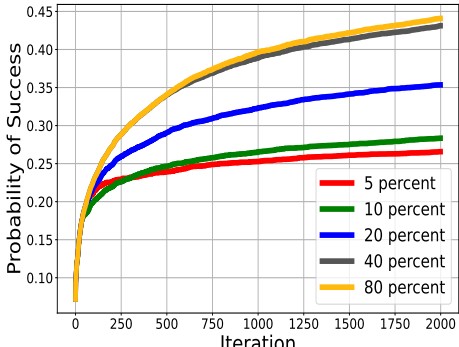

Figure 9: Hash-inversion accuracy increases with more optimization iterations and Larger hash sizes (Brand Table).

# E   Implications for Private Recommendation Systems

Our threat model is based on the common practices employed by the industry's recommendation systems. They are typically deployed in the cloud for inference serving Niu et al. (2020). In such a setting, a pre-trained model is hosted by a cloud server. The interaction history of each end user is kept in a user's local web browser or on a merchant's site where the merchant is precluded from sharing these data with other platforms without users' consent. This assumption is particularly important as it reflects the growing awareness in protecting personal data privacy.

There are various techniques that protect computations on cloud systems. These techniques include fully homomorphic encryption (FHE) Shmueli and Tassa (2017), multi-party computation (MPC) Goldreich (1998), and trusted execution environments (TEEs) Costan and Devadas (2016); Salter (2021). However, none of these techniques protect the privacy of memory access patterns. For example, while Intel SGX protects computational confidentiality and integrity, it has been shown to be vulnerable to side-channel attacks via memory access pattern leakage Wang et al. (2017). This paper shows that the information leakage through embedding table accesses may be used to extract private user information, suggesting that memory access patterns need to be protected if strong privacy protection is necessary for recommendation systems in the cloud.

Table 1 summarizes the attacks introduced in this paper. Each of them has a different goal. In all of these attacks, an attacker launches the attack by exploiting and analyzing the access patterns they observe. In some of the attacks, an attacker uses prior knowledge gleaned from the distribution of the accesses. In this work, we also define different metrics to evaluate each of these attacks. The high success rate of these attacks, highlights the importance of access pattern protection in the cloud-based recommendation systems.

# F   Related Work

The risk of information leakage in recommendation systems has been explored in prior works. However, most of the research in this area focused on other models (e.g. content filtering) or dense features. Access pattern privacy in recommendation systems is a new topic and current Federated learning and Oblivious RAM schemes have shortcomings when it comes to DNN-based recommendation systems as we discuss here.

The study in Zhang et al. (2021) designed a membership inference attack against a recommendation system to infer the training data in a content filtering model.Abdelberi *et al.* used a statistical learning model to find a connection between users' interests and the demographic information that users are not willing to share Chaabane et al. (2012). Previous studies also investigated the risk of cross-system information exposure Chaum (1985); Sweeney (2002). For instance, a former Massachusetts Governor was identified in voter registration records by the combination of a zip code, a birth date, and gender. Using this information, the researchers were able to identify him in a supposedly anonymous medical record dataset Sweeney (2002). Most of the prior research in this domain was focused on information leakage through dense features Akhtar and Mian (2018); Choquette-Choo et al. (2021); Li and Zhang (2021); Calandrino et al. (2011); Beigi and Liu (2020). Also, there are prior works investigating sparse feature leakage in other domains Ghinita et al. (2008); Aggarwal and Yu (2007). However, these leakages are through sparse feature values and not the embedding table accesses. Sparse feature's information leakage through embedding table accesses was explored for NLP models Song and Raghunathan (2020); Aggarwal and Yu (2007). This attack aimed to disclose the embedding tables' input values based on their output which is different from our threat model. Access pattern attacks are also investigated in databases research Grubbs et al. (2019); Bindschaedler et al. (2017). However, these attacks and defense schemes are fundamentally different from the ones in recommendation systems. In databases attack the goal is to find the value of the encrypted data of the database based on the range queries or the correlation of different rows.

Using federated learning for training centralized recommendation models has gained attention recently Yao et al. (2021); Yang et al. (2020). One of the problems of using federated learning for recommendation systems is the large size of embedding tables. These schemes usually use decomposition techniques such as tensor train to fit embedding tables on the edge devices Oseledets (2011). However, because of the accuracy drop, the compression ratio is not high which makes them incompatible with edge devices. TT-Rec mitigates the performance degradation of tensor decomposition by initializing weight tensors by Gaussian distribution Yin et al. (2021). Niu *et al.* proposed an FL framework to perform a secure federated sub-model training Niu et al. (2020). They employed Bloom filter, secure aggregation, and randomized response to protect users' private information. But, inference solutions are not discussed in these federate learning approaches. DeepRec Han et al. (2021) proposed an on-device recommendation model for RNNs. In this work, there is a global model trained by public data that is available from before GDPR. Each device downloads this global model and re-train the last layer with their data. The problem with this model is that it depends on before GDPR public data. However, with new models come new features, which were not collected before. Thus they can not rely on this scheme for future models.

