# OpenReview forum: "Private Data Leakage via Exploiting Access Patterns of Sparse Features in Deep Learning-based Recommendation Systems"
_NeurIPS.cc/2022/Workshop/TSRML — TSRML2022_

### Official Review · Reviewer_5ccr · 2022-10-19
**Interesting work, but threat model/assumptions need to be better clarified.**

**Overall Rating:** 4

**Summary:**

This paper demonstrates a privacy attack based on query patterns in recommender model lookup tables. The authors show several concrete ways that an attacker could identify a user or a user's private characteristics.

**Strengths:**

- This attack is novel and concerning. The authors clearly and succinctly explain the problem and potential consequences.

- I appreciated the way the authors separated the four attacks (sections 3-6) to make it clear what techniques could be used to infer different private characteristics.

**Weaknesses:**

- It's not clear what information the attacker does/does not have access to. Based on Figure 2, I would assume that the attacker can only see the access pattern to the embedding table. Yet later, in lines 150-151, it seems to say that an attacker could also know what product a user interacts with (e.g. Legoland) and could use this to make inferences. Additionally, at points it seems you assuming that the attacker knows what features are present in the embedding tables? This assumption is baked in, but it is never stated clearly. Please clarify this.

- In Section 4, the authors note (lines 152-153) that a "general attacker" could use their prior knowledge to link items to demographic groups. Obviously, you are able to do this in your attack, since you have the dataset and know the demographics. I'm not sure I see how this could happen in the real world. Does the general attacker have an additional source of information about products/features and general demographics of each? If so, what is it? This relates to the point above -- clarify your assumptions.

- This paper relates to ML insofar as it addresses a format of data commonly used for ML (recommender) models. The data privacy leakage observed is due to how the data is generated and processed before entering an ML system, not due to the ML model itself. While concerning, this seems distinct from the the workshop's goals, which is to "tackle open challenges on building trustworthy and socially responsible machine learning __models__". Private data processing is, of course, part of the pipeline, but this paper and its content seems better suited for a systems conference, not an ML one. Furthermore, the authors of this paper highlight a privacy attack without providing any potential solution. At the very least, they should hint at a solution.

- Minor points:
    - Reference format is strange and hard to read. Can you make these hyperlinks or otherwise differentiate them from the surrounding text?
    - Line 109 -- how did you compute that only 1120 combinations of features were possible? Is this specific to this dataset, or did you collapse features in some way?



**Overall Recommendation:**

I recommend rejecting this paper, as it is not yet ready for publication and also does not seem like a good fit for this workshop. I recommend that the authors address the threat model weaknesses I raised and consider a systems conference as a better fit.

**Review Confidence:**

3: The reviewer is fairly confident that the evaluation is correct

---

### Official Review · Reviewer_19Ag · 2022-10-20
**More like a system paper than a machine learning paper**

**Overall Rating:** 5

**Summary:**

In this paper, the authors discussed the private data leakage problem in certain circumstances. The authors present 4 attack methods and evaluate them with real-world datasets.

**Strengths:**

1 The authors explicitly describe the problem, and evaluation metrics.
2 The datasets are real-world datasets, and their methods are scalable to the massive data.



**Weaknesses:**

Weakness
1 For attack methods, the authors mainly use words to describe them, using more formulas will be clearer. For the result part, it will be better if the authors highlight their contribution.

2 in section 5, the authors need to claim the difference between the “re-identification attacks” and “identity resolution”, in these sections, it seems that the attackers are trying to do an identity resolution for the victims. If so, the authors should cite some identity resolution papers.

Minor:
Formula (9) goes up to the text.


**Overall Recommendation:**

Overall, the topic is interesting, but the quality of this paper needs to be improved.

**Review Confidence:**

4: The reviewer is confident but not absolutely certain that the evaluation is correct

---

### Decision · Program_Chairs · 2022-10-23

**Decision:**

Accept

**Comment:**

This paper studies private data leakage in a deployed machine learning system, and the attack exploits the access patterns to identify which entries of the sparse feature is nonzero. The attack focuses on the hardware side and is unclear for most machine learning reviewers. However, I still feel it is important to explore the hardware security aspect of machine learning in future works and this is a direction that the ML system community should look further.